# Relationship between social cognition and emotional markers and acoustic-verbal hallucination in youth with post-traumatic stress disorder: Protocol for a prospective, 2-year, longitudinal case-control study

**Louise-Emilie Dumas**[1,2]\*, **Arnaud Fernandez**[1,2,3], **Philippe Auby**[1,2], **Florence Askenazy**[1,2,3]

**1** University Department of Child and Adolescent Psychiatry, Pediatric Hospitals of Nice CHU-Lenval, Nice, France, **2** Université Côte d'Azur, CoBTek, FRIS, Nice, France, **3** Expert Center for Psychotrauma Paca Corse, Nice, France

\* louise-emilie.dumas@hpu.lenval.com

**Data Availability Statement:** No datasets were generated or analysed during the current study. All

## Abstract

### Introduction

Auditory-verbal hallucinatory experiences (AVH) have a 12% prevalence in the general pediatric population. Literature reports a higher risk of developing AVH in post-traumatic stress disorder (PTSD). The persistence of AVHs during adolescence represents a risk of evolution into psychotic disorders. Social cognition and emotional markers could be considered prodromes markers of this evolution. The objectives of this prospective observational study are to observe social cognition and emotional markers correlation with the presence and persistence of AVH over two years and with the evolution of PTSD and psychotic diagnosis.

### Methods and analysis

This prospective case-control study, longitudinal over two years (with an interim reassessment at six months and one year), will include 40 participants aged 8 to 16 years old with a diagnosis of PTSD and without a diagnosis of psychosis according to the criteria of DSM-5 (K-SADS-PL). Subjects included are divided into two groups with AVH and without AVH matched by gender, age and diagnosis. The primary outcome measure will be the correlation between social cognition and emotional makers and the presence of AVH in the PTSD pediatric population without psychotic disorders. The social cognition marker is assessed with the NEPSY II test. The emotional marker is assessed with the Differential Emotion Scale IV and the Revised Beliefs About Voices Questionnaire. The secondary outcome measures are the correlation of these markers with the persistence of AVH and the evolution of the patient's initial diagnosis two years later.

relevant data from this study will be made available upon study completion.

**Funding:** The author(s) received no specific funding for this work.

**Competing interests:** The authors have declared that no competing interests exist.

## Discussion

The originality of our protocol is to explore the potential progression to psychosis from PTSD by cognitive biases. This study supports the hypothesis of connections between PTSD and AVH through sensory, emotional and cognitive biases. It proposes a continuum model from PTSD to psychotic disorder due to impaired perception like AVH.

## Trial registration

**Clinical trial registration:** ClinicalTrials.gov Identifier: NCT03356028. https://classic.clinicaltrials.gov/ct2/show/NCT03356028.

## Introduction

Hallucination is an experience described as "a perception without an object to perceive" [1]. It can occur in the absence of organic disorders (metabolic, genetic, infectious, neurological or toxicological), and physiological events (such as hypnagogic or hypnopompic hallucinations) and differs from controlled imaginary productions [2]. These hallucinations as a developmental phenomenon [3] linked to psychic immaturity [4,5]. They can appear from the age of 5 years [6,7] with a prevalence in the general pediatric population of 17% in 9–12 years old and 7.5% in 13–18 years old [6,8]. They can evolve isolated, without other associated psychotic symptoms (delusions, disorganized thinking, behavioral disturbance) and without altering the child's relationship with reality and with their environment [9]. They are comorbid with psychic disorders like depression, anxiety and behavioral disorders [10,11], have a high risk of suicide [12,13] and represent a risk of impaired functioning of the child on a daily basis [14]. Literature has described multisensory hallucinations in the pediatric population that are more frequent [6,15] than in adults. Acoustic-verbal hallucination (AVH) is the sensorial modality chosen because it appears the most found in clinical practice [6] and is described in the literature with an identical prevalence in children and adolescents of 12% [3,16]. Hallucinatory experiences are found in childhood adversities or trauma, and trauma is a factor of hallucinations persistence as recurrent sensory memories [17,18]. The diagnosis of post-traumatic stress disorder (PTSD) includes symptoms of intrusion (repetitive memories), avoidance of stimuli associated with trauma, cognitive and mood alterations, and hypervigilance [19]. Intrusion symptoms such as visual or auditory revival in flashbacks may represent a risk for the development of AVH [20–23]. In most cases, hallucinations in children and adolescents are described as transient and benign [5,14]. Studies report that the persistence of AVH indicates a risk of evolution to a psychiatric disorder. Their persistence increases the risk of aggravation of mood disorders, anxiety and associated behaviors. The literature also points to the risk of evolution to schizophrenia spectrum disorder, especially if hallucinations persist into adolescence [4,24,25]. Social cognition disorders are considered prodromal markers, even indicators of a risk of evolution to schizophrenia spectrum disorder [26,27]. Impaired theory of mind and affect recognition abilities are described in the positive symptomatology of schizophrenia and contribute to disorder persistence and worsening [2,15]. Emotional dysregulation is a marker also described as promoting a distortion of reality [28], such as the hallucinatory experience.

A previous study in Nice was conducted in 2016 in CHU Lenval (Nice) to identify social cognition and emotional markers associated with AVH in non-psychotic children and adolescents. These previous results showed that social cognitions were not associated with the

presence or persistence of AVH, but emotional markers were associated with the persistence of AVH (i.e. fear, anger and sadness). It was also found that PTSD was significantly associated with the persistence of AVH [29].

We reproduce the protocol of the previous study to explore AVH in children and adolescents with PTSD and without any other sign of psychosis. We hypothesize that social cognition and emotional experiences are common markers to explain the presence and persistence of AVH in non-psychotic children and adolescents with PTSD. We based on the continuum model of Van Os [30–32], between infra-clinical and clinical signs of a psychotic phenotype. Like in children and adolescents with early onset schizophrenia studies, we supposed that social cognition and emotional markers [33–36] would act as a traumatic information processing bias [2] and would promote the presence and persistence of AVH. There is a high prevalence of traumatic antecedents or social adversity in the life of patients with hallucinatory phenomena in child and adult population studies [37]. Based on psychotic adult studies [38,39], Laroi and al. describe hallucination in children and adolescents as a defense mechanism against an intrusive cognitive event that engenders an unpleasant affect (distress, anxiety) [40]. Traumatic antecedents associated with impaired social cognition and emotional markers would cause a less effective compensatory strategies for managing anxiety, and make the subject vulnerable to the appearance of hallucinations [30,41]. If stressors may favor the presence of hallucinations, hallucinations can also maintain traumatic symptoms and promote the persistence of the trouble [42]. The hypothesis of our study is that social cognitions and emotional markers are expressed differently in the presence or absence of AVH in children and adolescents with PTSD. This study offers a prospective observation of these markers in the evolution of AVH and psychotic disorder in children and adolescents with PTSD.

## Objectives

The main objective of our study protocol is to compare social cognition markers and emotional markers according to the presence or absence of AVH in non-psychotic children and adolescents with PTSD. The secondary objectives of our study protocol are: 1) To compare social cognition markers and emotional markers according to the persistence at 2 years of AVH in non-psychotic children and adolescents with PTSD; 2) To evaluate the correlations between social cognition markers and emotional markers with AVH and with the persistence of AVH over two years; and 3) To evaluate social cognition and emotional markers associated with the evolution of PTSD and the development of psychotic disorder at two years.

## Methods and analysis

### Study design and sitting

Physalis study is a observational prospective, monocentric biomedical research protocol, on a case-control model and open, longitudinal over 2 years. Participants will be followed longitudinally in four-time assessment points: T0: baseline assessment; T1: at six months; T2: at one year; T3: at two years. It is conducted in the University Department of Child and Adolescent Psychiatry in Pediatric Hospitals of Nice CHU-Lenval. The time from first inclusion to study completion is expected to be seven years, including a recruitment phase (5 years) plus a follow-up (2 years).

### Study sample

Participants are children and adolescents including in- and out-patients of child psychiatric units with a diagnosis of PTSD (DSM-5).

*Inclusion criteria*: boys and girls aged 8 to 16; without intellectual deficit (QIT > 70); with a diagnosis of PTSD according to DSM-5 criteria (K-SADS-PL); receiving child psychiatry care for a diagnosis of mood, anxiety and/or behavioral disorder.

*Exclusion criteria*: a diagnosis of schizophrenia spectrum disorder according to DSM-5 criteria (K-SADS-PL); the presence of genetic, neurological, neurodevelopmental (including autism spectrum disorder) and neurosensory pathologies.

This case-control model observes a population with PTSD. Subjects included are divided into two groups: an experimental group with the presence of AVH ("AVH +" group), and a control group without AVH ("AVH–"group). Participants in the two groups will be matched by gender, age (more or less six months), and initial diagnosis based on DSM-5 criteria [19,43,44].

## Recruitment and consent

The recruitment is performed by psychiatrists at the University Department of Child and Adolescent Psychiatry of the Pediatric Hospitals of Nice—CHU Lenval. Patients are addressed to the study coordinators by the child psychiatrist in charge of the patient's care or selected through the electronic registries of the CHU Lenval. All participants are contacted individually by an experienced and trained evaluator. During the screening, patients are informed about the study procedures. There is no financial compensation for the study participation. Before being enrolled, subjects and their legal representatives must provide signed informed assents.

## Assessment and outcomes measures

**Assessment of inclusion and non-inclusion criteria.**    The PTSD and psychosis section of the psychometric test Kiddie Schedule for Affective Disorders and Schizophrenia for School-Age Children-Present and Lifetime (K-SADS-PL) [19]in the presence of the child and their parents, is employed to validate the existence of PTSD and the absence of psychotic disorders. K-SADS-PL is a semi-structured diagnostic interview to evaluate current and past episodes of psychopathology in children and adolescents based on DSM-5 criteria, validated in french version [44].

The absence of mental retardation is determined using the abbreviated form of the Wechsler Intelligence Scale for Children (WISC) IV. The french version of abbreviated form of WISC IV [45] permits the rapid passing of WISC IV tests while ensuring an accurate and reliable estimate of the Total Intelligence Quotient (TIQ).

The psychometric test Mini International Neuropsychiatric Interview Enfants-Adolescents (MINI-Kid) [46], validated in french version [47] explores in a standardized way the main psychiatric disorders of the DSM-5 axis in children. This interview is performed with the child and their parents. The presence of any initial psychiatric disorder is categorized using the MINI-Kid: mood disorder: (depression and (hypo)mania); anxiety disorder (panic disorder, agoraphobia, social phobia, obsessive-compulsive disorder and generalized anxiety); behavioral disorder (conduct disorder and attention deficit hyperactivity disorder (ADHD)).

All of these instruments were previously translated and validated in French.

A sociodemographic and clinical questionnaire, used in a previous study [29], is employed to collect information on perinatal, medical, surgical, psychological and psychiatric history, psychomotor development, family history, biographical information, and family environment. A specific part focuses on hallucinatory phenomena to clarify them, to exclude hypnopompic and hypnagogic episodes, imaginary productions and companions. We then establish a more detailed description of hallucinations by analyzing their characteristics: age of onset, clinical type, content, frequency, location tone and number of voices heard.

**Assessment for group allocation.** AVH test questionnaire, used in a previous study [29], is employed to classify participants in "AVH +" or "AVH–"groups. This questionnaire is inspired by the schizophrenia section of the Diagnostic Interview Schedule for Children-Child (DISC-C) [48,49], two of which relate to the existence of AVH. The study evaluator posed the following question to all participants included in the study: "Have you ever heard a voice in your head that is different from your own and that no one else hears except you?" to accurately determine whether they would belong to the "AVH+" or "AVH-"group.

## Outcomes measured

- *Primary outcome*

Social cognition skills are assessed using NEPSY II's "Theory of Mind" and "Emotion Recognition" tests [50], in french version [51]. The "Theory of Mind" test assesses the child's ability to comprehend other perspectives, intentions, and beliefs. "Emotion Recognition" assesses facial affect recognition ability based on six facial expressions: joy, sadness, anger, fear, disgust and a neutral expression.

Participants' emotional experience is assessed using the Differential Emotional Scale IV (DES IV) [52], in french version [53]. This self-report scale measures the subjective emotional experience by assessing the experience of the following: interest, joy, surprise, anger, contempt, disgust, sadness, fear, guilt, shame, shyness, and hostility against self. This instrument measures the emotions-traits representing stable individual differences, as expressed by the frequency with which emotions are experienced in everyday life.

The Beliefs About Voices Questionnaire-Revised (BAVQ-R) [54,55]is a self-report scale that characterizes patients' relationships with their hallucinations, validated in french version [56]. It determines the omnipotent, benevolent or malicious nature of the voices. It also explores the subject's reaction to their voices, either the ability to resist or the decision to follow what they say.

- *Secondary outcome*

The evaluation of the persistence of non-psychotic AVH is made using the same AVH test questionnaire used during the inclusion phase.

The assessment of the psychiatric diagnosis is made using PTSD and psychosis section of the K-SADS-PL and the MINI-Kid.

## Procedure

This study includes two stages: the screening and study assessment performed immediately after inclusion (T0), and a follow-up assessment performed after six months (T1), one year (T2) and two years (T3).

These study procedures are illustrated in Figs 1 and 2.

The study model does not allow blinding because it is a case-control study and the patient's clinical data contains the question of the presence or absence of AVH at each stage of the protocol. During this study, participants continue their usual child psychiatric follow-up, and other types of treatment can be offered.

## Sample size

The current literature data does not provide enough information for the calculation of the effect size and the required sample size. We were consequently unable to calculate a number of subjects necessary for the physalis study. We calculated a number of necessary subjects on the

|  | STUDY PERIOD | | | | | |
|---|---|---|---|---|---|---|
|  | **Enrolment** | **Allocation** | **Post-allocation** | | | **Close-out** |
| **TIMEPOINT** | T0 | T0 | T0 | T1 | T2 | T3 |
| **ENROLMENT** | | | | | | |
| **Eligibility screen** PTSD and psychosis section of K-SADS-PL | X | | | X | | X |
| Abbreviation form of WISC IV | X | | | | | |
| **Informed consent** | X | | | | | |
| **Allocation** AVH test questionnaire | | X | | X | X | X |
| **INTERVENTIONS** | | | | | | |
| Interview patients and parents | | | X | X | | X |
| Phone call | | | | | X | |
| **ASSESSMENTS** | | | | | | |
| **Baseline variable:** Social-demographic and hallucinations database | | | X | | | |
| **Outcome variables :** NEPSY II DES IV BAVQ-R | | | X | | | X |
| **Other data variable:** MINI-Kid | | | X | X | | X |

**Fig 1. SPIRIT Schedule of enrolment, interventions and assessments.**

feasibility of recruiting and monitoring patients in the child psychiatry department and based on our previous work: a number of 15 patients per group, and a total number of 30 patients in total, were chosen. Considering a rate of loss of follow-up and missing data of around 30%, 20 patients are to be included per arm for a total of 40 patients.

## Data analysis

The statistical analysis includes a descriptive study of the series of absolute and relative frequencies for categorical variables, and the estimation of median and interquartile mean and standard deviations for the quantitative variables.

- *The main objective*

In order to answer the study hypothesis, a bivariate descriptive analysis will look for a significant difference between the AVH+ and AVH- groups. Continuous variables compared

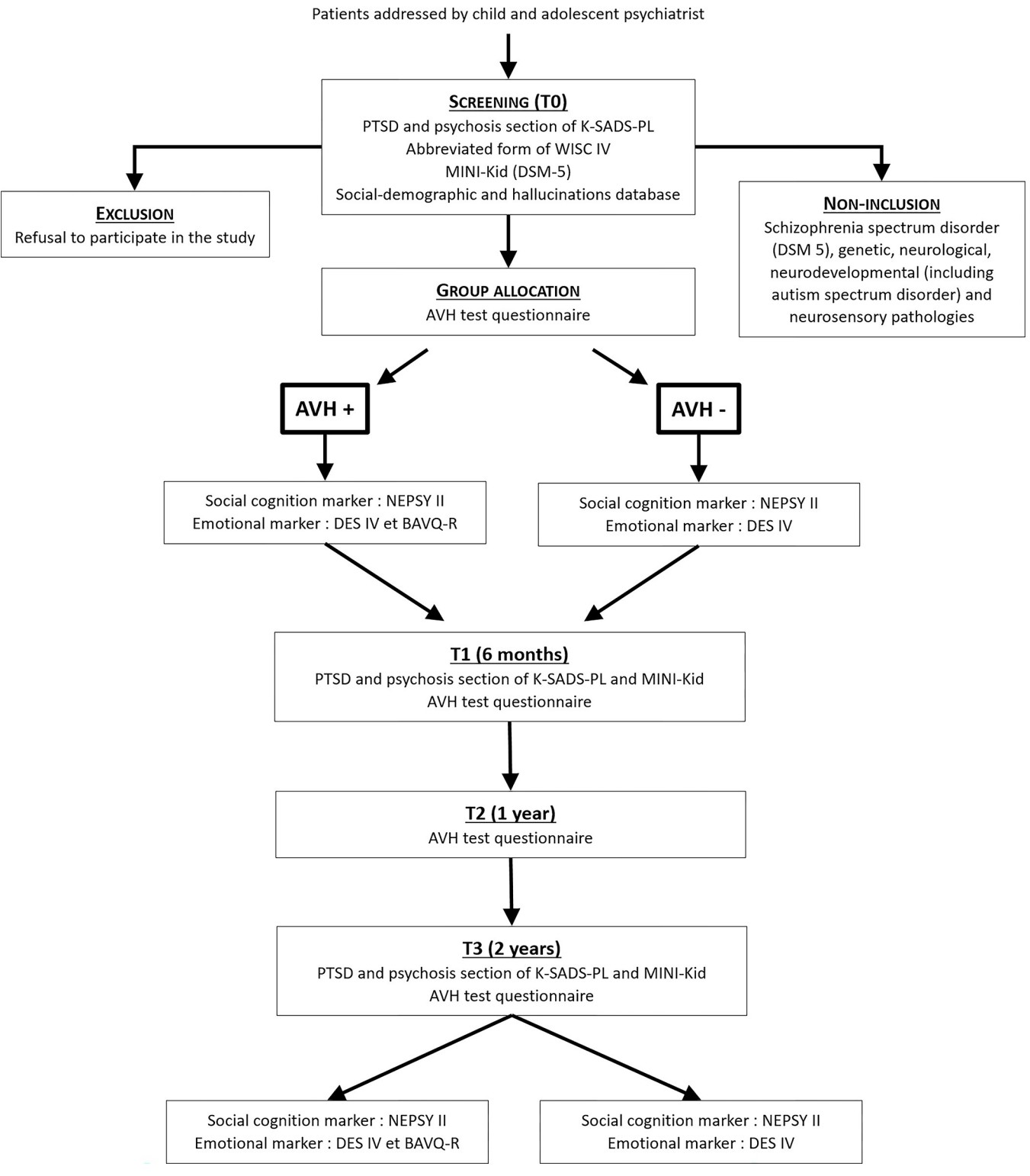

**Fig 2. Flow chart.**

using the Student test (or the non-parametric Wilcoxon test if the variables do not follow a normal distribution) concern the emotional markers "DES IV" and "BAVQ-R". Categorical variables compared using the chi-square test (or the non-parametric two-tailed Fisher exact test if the variables do not follow a normal distribution) concern the social cognition markers NEPSY II "Theory of Mind" and "Emotion Recognition ". A Cramer's V test in addition to the chi-square test will be carried out in order to measure the intensity of the relationship of nominal variables between two groups. Since there are two primary outcomes, correcting the alpha risk by the Bonferroni method is necessary, with an adjusted alpha of 2.5% for each marker to control the overall familywise error rate.

- *Exploratory analyses*

  1. A new statistical analysis will be carried out at two years to compare social cognition markers and emotional markers between "AVH+" and "AVH-" groups.

  2. The correlation between the "AVH +" and "AVH -" groups for social cognition (NEPSY II) and emotional markers (DES IV and BAVQ-R) in non-psychotic children and adolescents will be evaluated using correlation analyses. Pearson's correlation method was used to measure the association between two continuous variables in a small population sample. For the correlation of categorical variables, the nullity test of a correlation coefficient will be used (or the Spearman method in the case where the variables do not follow a normal distribution). Correlations between social cognition and emotional markers and the persistence of AVH in non-psychotic children and adolescents at two years will be reassessed.

  3. The variables of the groups with or without persistence of AVH at two years will be compared and correlated to evaluate the evolution of the initial diagnosis of PTSD and the development of a psychotic disorder.

The R statistical software was used in this study [57].

## Status of the study

The inclusion of participants in the study began on February 8, 2020. The number of inclusions necessary was reached on February 8, 2023. The participants in the study are being reassessed. The end of participation in the study for the last participants included will be on February 8, 2025.

## Ethics and dissemination

### Informed consent and institutional review boards

The study is based on the principles of Good Clinical Practice (GCP), according to the Declaration of Helsinki. Patients and both their parents or legal representative receive oral and written explanations of the study including its potential risks, their right to withdraw consent at any time and the details of data protection and confidentiality; sufficient time to ask and answer questions is guaranteed. A signed consent form is obtained. The patient's information and a copy of the signed consent form are handed to the patient.

The protocol of the physalis study is ancillary to the protocol "Child and adolescent psychiatry and multidisciplinary research (public health, psychodynamics, neurosciences and human and social sciences) with children exposed to the Nice attack of 14 July 2016 (Program 14–7)". Program 14–7 has the reference number 17-HPNCL-03 and was approved by the Ethics Committee (CE) "Nord-Ouest III" (France) on July 13, 2017 and the National Agency for the Safety

of Medicines and Health Products and registered with the National Medicines Safety Agency (ANSM) and all procedures will comply with ethical guidelines international ethics for clinical studies involving human beings. The study protocol has been registered on Clinical Trials.gov (Identifier:NCT03356028).

The drafting of the physalis study protocol began in 2018. As the protocol targets patients with PTSD, we moved closer to ongoing studies implicating patients in the attacks of July 14, 2016. An amendment was requested to include the physalis study as an ancillary to program 14–7 and approved by the Ethics Committee (CE) "Nord-Ouest III" (France) on July 27, 2019. In order to facilitate patient inclusion, we have opened the inclusion criteria patients with PTSD to any type of trauma. A new request for amendment was made and approved by the Ethics Committee (EC) "Nord-Ouest III" (France) on February 8, 2020.

## Confidentiality

All documents and information are treated with strict confidentiality. The information collected in the study, especially the information related to the patient's identity, is confidential and protected by law. The data collected at each study site is declared to the "Commission Nationale de l'Informatique et des Libertés" (CNIL) stored and analyzed in anonymized form, and kept for a period of 15 years in a lockable cabinet and a password-protected computer. The collected data will be only accessible to the principal investigator and study staff.

## Dissemination

After study completion, the results of the primary and secondary analyses will be published in international peer-reviewed journals. If shown to provide valid and reliable information in addition to traditional measures, findings would be presented in international symposiums to consider ways to develop new research protocols on AVH in non-psychotic children.

## Discussion

### Clinical implications

In the pediatric population, previous studies have described the characteristics of the hallucinations [58,59], the age of onset [60–62], the comorbidity [63–69] and the evolution [60,70–72] and explored cognitive bias in the evolution of psychosis [26,27,73,74]. The literature demonstrates that the persistence of hallucinations promotes the development of psychosis [60]. The link between childhood trauma and clinical high-risk psychosis [75,76] due to cognitive bias [77,78] is also reported.

In line with the literature, we hypothesized connections between PTSD and AVH through sensory, emotional and cognitive biases [79]. In this study, we assume the existence of a continuum from PTSD to psychotic disorder due to impaired perception like AVH. These biases could promote the persistence of AVH and favor the evolution to psychotic disorders.

Cognitive and behavioral models of the etiology and maintenance of hallucinations take into account the symptoms of PTSD. They suggest that post-traumatic intrusions play an essential role in the development of hallucinations [80]. In PTSD, hypervigilance excites the sensory arousal involved in flashbacks [81]. Cognitive biases of causal attribution lead to a misinterpretation of these post-traumatic intrusions and favor the development of hallucinations [82]. Conversely, hallucinations are poorly contextualized sensory information that inadvertently recalls the trauma [83]. The model acts as a retroactive loop which fixes the

symptomatology and maintains the psychic suffering. Hallucinations cause the subject to relive the negative emotions experienced during the trauma: in our previous study, the emotional marker associated with the persistence of AVH was negative emotions such as sadness, fear, anger and hostility [29].

Delusional thinking arises from searching for an explanation for these intrusions, which is influenced by underlying traumatic beliefs about self and others, as well as negative emotions. Social cognition mediates the attribution of sensory intrusions and trauma-related beliefs that the self is vulnerable and others are hostile producing delusions (particularly paranoia) and maintaining hallucinations [80].

All of these previous works retrospectively studied PTSD in psychotic symptoms [78]. The originality of our protocol is to explore the potential progression to psychosis from PTSD by cognitive biases. We suggest that the disturbance of sensory cognition in PTSD (post-traumatic intrusion), could be a traumatic "dys-sensory perception". This "dys-sensory perception" would be the starting point that cascades negative emotions and reinforces the bias of social cognition in beliefs related to trauma. Understanding the interplay between trauma-related psychological mechanisms and psychotic symptoms may improve the effectiveness of interventions for post-traumatic stress reactions in psychosis [81].

## Limitations

The principal limitation of our protocol is the small sample size of the study population recruited over a relatively long period, i.e. two years. The small population sample of the study is due to feasibility. The presence of auditory verbal hallucinations (AVH) in children and adolescents with post-traumatic stress disorder (PTSD) is a rarer symptomatology and more challenging to objectify in clinical interviews compared to the adult population. Previous studies have shown that only 1 to 30% of parents would be informed about hallucinations of their children [64] and only a minority of psychiatrists are aware of these symptoms [64,84,85]. Despite our child and adolescent psychiatry department being a reference center for psychotrauma, the number of patients exhibiting this clinical profile remains limited. The Pearson correlation method to measure the association between two variables was chosen rather than a multiple regression method given the number of subjects was too low compared to the number of variables of the study. The choice of this method for a small sample size remains a limitation in the analysis of the results. The question used at patient inclusion to determine their assignment to the "AVH+" or "AVH-"group was not a standardized tool and could introduce a classification bias. The only evaluator of the study deliberately and consistently posed this question to all participants included in the study to ensure the reproducibility of the test. Despite these limitations, the originality of this topic and the clear clinical interest could highlight the importance of confirming these results in a more extensive study.

## Supporting information

**S1 Checklist. SPIRIT 2013 checklist: Recommended items to address in a clinical trial protocol and related documents\*.**
(DOC)

**S1 File.**
(PDF)

**S2 File.**
(PDF)

## Acknowledgments

The authors would like to acknowledge the contributors of the study and Foundation Lenval, Pediatric Hospitals of Nice CHU-Lenval.

## Author Contributions

**Conceptualization:** Louise-Emilie Dumas, Florence Askenazy.

**Investigation:** Louise-Emilie Dumas.

**Methodology:** Louise-Emilie Dumas.

**Software:** Louise-Emilie Dumas.

**Supervision:** Arnaud Fernandez, Florence Askenazy.

**Writing – original draft:** Louise-Emilie Dumas.

**Writing – review & editing:** Arnaud Fernandez, Philippe Auby, Florence Askenazy.

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
