## [Decision Letter · Decision Letter 0]

7 Nov 2023

PONE-D-23-25826Relationship between social cognition and emotional markers and acoustic-verbal hallucination in youth with post-traumatic stress disorder: protocol for a prospective, 2-year, longitudinal case-control studyPLOS ONE

Dear Dr. Dumas,

Thank you for submitting your manuscript to PLOS ONE. After careful consideration, we feel that it has merit but does not fully meet PLOS ONE’s publication criteria as it currently stands. Therefore, we invite you to submit a revised version of the manuscript that addresses the points raised during the review process.

We look forward to receiving your revised manuscript.

Kind regards,

Tord Ivarsson, MD, PhD

Academic Editor

PLOS ONE

Journal Requirements:

2. "In your Data Availability statement, you have not specified where the minimal data set underlying the results described in your manuscript can be found. PLOS defines a study's minimal data set as the underlying data used to reach the conclusions drawn in the manuscript and any additional data required to replicate the reported study findings in their entirety. All PLOS journals require that the minimal data set be made fully available. For more information about our data policy, please see http://journals.plos.org/plosone/s/data-availability.

3. We note that the original protocol that you have uploaded as a Supporting Information file contains an institutional logo. As this logo is likely copyrighted, we ask that you please remove it from this file and upload an updated version upon resubmission.

Additional Editor Comments:

Dear Dr Dumas,

PLOS aplogizes for the long time this article has been under review, which largely is due to difficulties to get reviews.

I recommend you to make a "major revision" as our two reviewers had some serious concerns regarding the protocol.

However, as the protocol also has some very clear strengths, I ask you to consider carefully the points made by the reviewers and then revise your submission thoroughly.

If you consider that some points raised by the reviewers go too far, or not correct, your rebuttal of each point must be very clear and cogent.

First, the issue of ethics raised by needs to be sorted out, as proper permission is central to publication.

Second, the power issues is important too, because the collection of data that are bound to become indecisive with regard to the research question is problematic as well. Twenty cases in each group are quite few, and as a longitidinal design is fraught with attrition, power issues can not be taken lightly. Furthermore, if you have solid grounds for believing that 20 cases in each group will be enough, this pre-supposes a power analysis, in which case it needs to be included in the re-submission.

Third, the analysis plan was criticized. You state that you will use two years observation time, and as the research question implicates change across time, this must be clarified in the analysis plan.

As academic editor I agree with the reviewers independently through my own reading, and I hope that you will be able to revise and re-submit.

Kind regards,

Tord Ivarsson

Reviewers' comments:

Reviewer's Responses to Questions

**Comments to the Author**

1. Does the manuscript provide a valid rationale for the proposed study, with clearly identified and justified research questions?

Reviewer #1: Yes

Reviewer #2: Partly

2. Is the protocol technically sound and planned in a manner that will lead to a meaningful outcome and allow testing the stated hypotheses?

Reviewer #1: No

Reviewer #2: Partly

3. Is the methodology feasible and described in sufficient detail to allow the work to be replicable?

Reviewer #1: No

Reviewer #2: Yes

4. Have the authors described where all data underlying the findings will be made available when the study is complete?

Reviewer #1: No

Reviewer #2: Yes

5. Is the manuscript presented in an intelligible fashion and written in standard English?

Reviewer #1: Yes

Reviewer #2: Yes

6. Review Comments to the Author

You may also provide optional suggestions and comments to authors that they might find helpful in planning their study.

Reviewer #1: This protocol refers to an interesting and important topic, namely the trajectory of hallucination and psychosis development in adolescents with PTSD. The authors aim to make use of a prospective design to eliminate issues that are currently present in the literature, namely the exclusive use of cross-sectional data. If completed, this study would contribute significantly to elucidating the relationship between PTSD, auditory hallucinations and psychosis development.

While relevant and interesting in scope, the study protocol presents with a number of issues that I think are necessary to address:

1. Ethics approval/Trial registration: The proposed study is based on a pilot study which was registered under clinicaltrials.gov (https://clinicaltrials.gov/study/NCT02567500#publications). However, the ethic amendment for the study protocol is related to another study investigating the effects of exposure to the attack in Nice on July 14th, 2016 (Program 14-7 Study https://clinicaltrials.gov/study/NCT03356028). Since the two studies (proposed (“Physalis”) and Program 14-7) are very different in terms of recruitment (14-7 is already recruited, specifically from people exposed to the attacks) it is not clear to me how these two studies are related. I would suggest the authors make clear in the manuscript; in what way the proposed study is an extension of the original study and why the pilot “Physalis” study had their own clinical trial entry, but this current extension has not.

In addition, the clinical trial reference to the program 14-7 study is not helpful, insofar as the recruitment strategy is very different from the proposed study. Furthermore, the amendment and attached study protocol mention the distinction between mass and individual trauma which does not occur in the current manuscript. If they proposed study does not recruit from the Program 14-7 sample at all, this should be made clear and in this case I do no think the current study can be considered an extension of the Program 14-7 study.

2. Study instruments: One of the main variables of interest in this study is the presence of auditory-verbal hallucinations. The authors briefly mention that the instrument was published previously (in a French study, so not accessible to international readers) but give no more detail. The authors should describe this instrument here in detail as well, especially since the outcome variables of this instrument (presence yes/no or a continuous outcome?) is important for the statistical analysis.

3. Sample size calculation: The authors mention that it is impossible to make a quantified hypothesis for the sample analysis, but then go on to provide some information on an analysis anyway. The analysis is missing the effect size that was considered for the calculation. The authors should either state clearly that they have not done a sample size analysis or perform sufficient detail to replicate this analysis (see also the Work of Daniel Lakens regarding sample size justification https://psyarxiv.com/9d3yf/). Additionally, the authors mention the small sample size in their limitations section, however the study is still running. Why not recruit more patients if that’s the main limitation?

4. Data analysis: The data analysis plan is currently incomplete. The authors should mention the specific variables (MINI-KID; NEPSY, etc.) used in each analysis and give more analysis regarding the time point of each analysis. As of now the plan ready as if conducted in a cross-sectional study and it is not clear how the prospective design is considered in the analysis plan. Please include specific analyses regarding the time frame, and mention which analysis is done at which time point (T0, T1, etc.). Additionally, the main analysis seems to be a correlational analysis, however it is not clear to me that all involved variables continuous (especially since the AVH variable is not clearly described in the instruments sections). I suspect a between-group analysis (AVH+ vs. AVH-) would be more appropriate. A correlational analysis also seems not best suited for the relationship between patients’ diagnosis (categorical I assume) and AVH (again, probably a categorical variable). Here as well, the authors state that correlational analysis is used because of the small sample size, but since this is a study protocol, they can just plan to include more participants if that’s what needed for an appropriate analysis. Overall, this section is most concerning and needs the most comprehensive rewrite.

5. Hypothesis: In the discussion section the authors discuss a very interesting theory regarding the co-development of AVH and PTSD symptoms (“dys-sensory perception”), however as far as I can tell they plan no tests or analyses to investigate this model. I suggest the authors deduce specific predictions from this model that can be tested with their collected data.

Minor concerns:

1. Figure quality: The attached figures are not of sufficient quality for publication.

2. Hypotheses: The hypotheses in the introduction should be stated more clearly and in statistical terms.

3. Data availability: The authors should add a data availability statement to the method section, detailing where their data will be made available.

Reviewer #2: For the sample size calculation, the information that was used to derive the sample size is to be provided e.g. reference of the previous study is to be cited, and whether the sample size calculation in the previous study was performed.

The statistical software, its version, publisher name and the accepted level of significance is to be stated.

The language version for all the tools/inventories/questionnaires including validation information (where applicable) is to be stated.

The sentence ‘A bivariate descriptive analysis will look for a significant difference between the case and control groups by the Student's test for the comparison of quantitative variables (or the non-parametric test of Wilcoxon if the validity conditions are not validated), and the chi² test (or Fisher's exact test in the case where the validity conditions are not validated) ‘ requires revision.

For Fisher’s exact test, 1 or 2 -tailed test is to be stated. Chi2 test to be written as chi-square test.

The statement ‘Pearson's correlation method was used to measure the association between two variables in a small population sample.’ requires revision. Pearson's correlation can be used in small population samples to measure the association between two continuous variables but it is crucial to consider the nature of the data.

Typo 'Bonferonni' (Bonferroni)

The statement ‘Since these are two primary outcomes, correcting the alpha risk by the Bonferonni method is necessary, with alpha = 2.5% for each marker’ could be revised to ‘Since there are two primary outcomes, correcting the alpha risk by the Bonferroni method is necessary, with an adjusted alpha of 2.5% for each marker to control the overall familywise error rate.’

Some references did not conform to the journal format.

The authors have stated that the limitation of the study is due to the low sample size. This is a good and interesting study and since it is a prospective case-control study, it is a waste not to make an attempt to get more subjects in order to derive more meaningful data/results. Studying the relationship between social cognition and emotional markers and acoustic verbal hallucination in youth with post-traumatic stress disorder requires a good number of subjects. Also, there are many variables that could possibly be involved/interplay, where confounding, mediating, and moderating effects could present in the study and require more meaningful statistical analysis. The authors could make an attempt to increase the sample size by recruiting subjects from other sites. Also, the attrition rates could be more than what the authors have anticipated.

7. PLOS authors have the option to publish the peer review history of their article (what does this mean?). If published, this will include your full peer review and any attached files.

Reviewer #1: **Yes: **Dr. Lukas A. Basedow

Reviewer #2: No

---

## [Author Response · Author response to Decision Letter 0]

19 Dec 2023

Reviewer #1: This protocol refers to an interesting and important topic, namely the trajectory of hallucination and psychosis development in adolescents with PTSD. The authors aim to make use of a prospective design to eliminate issues that are currently present in the literature, namely the exclusive use of cross-sectional data. If completed, this study would contribute significantly to elucidating the relationship between PTSD, auditory hallucinations and psychosis development.

While relevant and interesting in scope, the study protocol presents with a number of issues that I think are necessary to address:

1. Ethics approval/Trial registration: The proposed study is based on a pilot study which was registered under clinicaltrials.gov Since the two studies (proposed (“Physalis”) and Program 14-7) are very different in terms of recruitment (14-7 is already recruited, specifically from people exposed to the attacks) it is not clear to me how these two studies are related. I would suggest the authors make clear in the manuscript; in what way the proposed study is an extension of the original study and why the pilot “Physalis” study had their own clinical trial entry, but this current extension has not.

In addition, the clinical trial reference to the program 14-7 study is not helpful, insofar as the recruitment strategy is very different from the proposed study. Furthermore, the amendment and attached study protocol mention the distinction between mass and individual trauma which does not occur in the current manuscript. If they proposed study does not recruit from the Program 14-7 sample at all, this should be made clear and in this case I do not think the current study can be considered an extension of the Program 14-7 study.

I thank you for your comment which allows me to provide clarification on the “ethical approvals/trial registration” paragraphs. This was to optimize recruitment possibilities for the physalis study. At the beginning of the Physalis study we thought that the 14-7 program would allow for easy recruitment to the Physalis study. Then after a few months, we concluded that it was necessary to open inclusions to all types of trauma and recruitment methods to achieve the number of patients necessary for the study. From February 8, 2020, patients were recruited in the context of the mass trauma of July 14, 2016 but also in the event of individual trauma only.

On a clinical level, the data in the literature do not seem to demonstrate that the type of trauma (mass or individual) can have an influence on the hallucinatory experience.

I replaced part of the paragraph “Informed Consent and Institutional Review Boards” : “The protocol of the physalis study is ancillary to the protocol "Child and adolescent psychiatry and multidisciplinary research (public health, psychodynamics, neurosciences and human and social sciences) with children exposed to the Nice attack of 14 July 2016 (Program 14-7)". Program 14-7 has the reference number 17-HPNCL-03 and was approved by the Ethics Committee (CE) “Nord-Ouest III” (France) on July 13, 2017 and the National Agency for the Safety of Medicines and Health Products and registered with the National Medicines Safety Agency (ANSM) and all procedures will comply with ethical guidelines international ethics for clinical studies involving human beings. The study protocol has been registered on Clinical Trials.gov (Identifier:NCT03356028).

The drafting of the physalis study protocol began in 2018. As the protocol targets patients with PTSD, we moved closer to ongoing studies implicating patients in the attacks of July 14, 2016. An amendment was requested to include the physalis study as an ancillary to program 14-7 and approved by the Ethics Committee (CE) “Nord-Ouest III” (France) on July 27, 2019. In order to facilitate patient inclusion, we have opened the inclusion criteria patients with PTSD to any type of trauma. A new request for amendment was made and approved by the Ethics Committee (EC) “Nord-Ouest III” (France) on February 8, 2020.”

2. Study instruments: One of the main variables of interest in this study is the presence of auditory-verbal hallucinations. The authors briefly mention that the instrument was published previously (in a French study, so not accessible to international readers) but give no more detail. The authors should describe this instrument here in detail as well, especially since the outcome variables of this instrument (presence yes/no or a continuous outcome?) is important for the statistical analysis.

Thank you for your comment, which allows us to specify the tool used in the study. The question, "Have you ever heard a voice in your head that is different from your own and that no one else hears except you?" is drawn from the schizophrenia section of the Diagnostic Interview Schedule for Children (Costello, 1984). The question used in the Physalis study is not part of a standardized tool. The only evaluator of the study deliberately and consistently posed this question to all participants included in the study to ensure the reproducibility of the test. However, it is important to note that this approach may introduce a potential bias into the study, and we will address this in the "Limitations" section of the article.

I added the following sentence in the 'Methods and Analysis' section, subsection 'Assessment and Outcomes Measures,' 'Assessment for Group Allocation': “The study evaluator posed the following question to all participants included in the study: “Have you ever heard a voice in your head that is different from your own and that no one else hears except you?” to accurately determine whether they would belong to the “AVH+” or “AVH-“ group."

I have modified the references associated with the mention of the 'Diagnostic Interview Schedule for Children-Child (DISC-C)' tool.

I also added in “Limitations” the following sentences: “The question used at patient inclusion to determine their assignment to the “AVH+” or “AVH-“ group was not a standardized tool and could introduce a classification bias. The only evaluator of the study deliberately and consistently posed this question to all participants included in the study to ensure the reproducibility of the test”.

3. Sample size calculation: The authors mention that it is impossible to make a quantified hypothesis for the sample analysis, but then go on to provide some information on an analysis anyway. The analysis is missing the effect size that was considered for the calculation. The authors should either state clearly that they have not done a sample size analysis or perform sufficient detail to replicate this analysis (see also the Work of Daniel Lakens regarding sample size justification https://smex-ctp.trendmicro.com:443/wis/clicktime/v1/query?url=https%3a%2f%2fpsyarxiv.com%2f9d3yf%2f&umid=7f3ccb42-b18d-45dc-83dc-5ed85ae75f98&auth=485ed4c97c395527e0d3511589861affbf66eedc-51aecf1c954ca848635a24732b1360850b6ef35d). Additionally, the authors mention the small sample size in their limitations section, however the study is still running. Why not recruit more patients if that’s the main limitation?

Thank you for this remark which allows me to correct the paragraph "Sample size" with the following sentence: “Because of the absence of reliable quantified data in the literature, it is impossible to make a quantified hypothesis for the variability of the endpoint. We were consequently unable to calculate a number of subjects necessary for the physalis study”.

I have also provided additional information in the "Limitations" paragraph with the following sentences modified: “The small population sample of the study is due to feasibility. The presence of auditory verbal hallucinations (AVH) in children and adolescents with post-traumatic stress disorder (PTSD) is a rarer symptomatology and more challenging to objectify in clinical interviews compared to the adult population. Previous studies have shown that only 1 to 30% of parents would be informed about hallucinations of their children (60) and only a minority of psychiatrists are aware of these symptoms (60,80,81). Despite our child and adolescent psychiatry department being a reference center for psychotrauma, the number of patients exhibiting this clinical profile remains limited”.

4. Data analysis: The data analysis plan is currently incomplete. The authors should mention the specific variables (MINI-KID; NEPSY, etc.) used in each analysis and give more analysis regarding the time point of each analysis. As of now the plan ready as if conducted in a cross-sectional study and it is not clear how the prospective design is considered in the analysis plan. Please include specific analyses regarding the time frame, and mention which analysis is done at which time point (T0, T1, etc.). 

Thank you for your comment, however the data analysis plan is illustrated in Figures 1 and 2 of the article (provided in another document in the submission procedure). These figures precisely describe which analyzes are carried out at each time of the study. But I can also add this in the text of the article with the following paragraph:

• “At T0, all patients:

- sign a consent to participate in the study after having received informed information,

- pass the questionnaires determining the inclusion criteria in the study: PTSD and psychosis section of K-SADS-PL and the abbreviation form of WISC IV,

- are asked about the presence or absence of AVH to determine their group assignment,

- respond to an interview concerning their socio-demographic data and hallucinatory symptoms,

- respond to the study variables using the NEPSY II, DES IV, BAVQ-R (only for the “AVH +” group) and MINI-Kid questionnaires.

• At T1, all patients are reviewed to assess the presence or absence of AVH as well as the evolution of PTSD and psychosis diagnoses (K-SADS-PL) as well as other diagnoses by the MINI-Kid.

• At T2, all patients are called by telephone to ask them only about the presence or absence of AVH.

• At T3, all patients are reviewed for:

- evaluate the evolution of their diagnosis by passing the PTSD and psychosis section of the K-SADS-PL and Mini-Kid,

- assess the presence or absence of AVH,

- retake the specific study questionnaires: NEPSY II, DES IV and BAVQ-R (if AVH is present).”

Additionally, the main analysis seems to be a correlational analysis, however it is not clear to me that all involved variables continuous (especially since the AVH variable is not clearly described in the instruments sections). 

I have included in the appendix of the article a table summarizing all the variables of the study.

Appendix: Description of study variables

Variable type Variable name Variable measurement

Continuous quantitative variables Age years

 Abbreviation form of WISC IV

Total Intelligence Quotient Point

 Differential Emotional Scale IV (DES IV):

Interest, Joy, Surprise, Anger, Contempt, Disgust, Sadness, Fear, Guilt, Shame, Shyness, and Hostility against self From 3 to 15 points/items

 The Beliefs About Voices Questionnaire-Revised (BAVQ-R):

- Omnipotence, 

- Benevolence, 

- Malevolence, 

- Emotional resistance, Behavioral resistance,

- Emotional engagement, Behavioral engagement From 4 to 24 points/items

Categorical variables Sex Boy (0) / Girl (1)

 AVH No (0) / Yes (1)

 PTSD (K-SADS-PL) No (0) / Yes (1)

 Psychosis (K-SADS-PL) No (0) / Yes (1)

 MINI-Kid

- mood disorder: (depression and (hypo)mania); 

- anxiety disorder (panic disorder, agoraphobia, social phobia, obsessive-compulsive disorder and generalized anxiety); 

- behavioral disorder (conduct disorder and attention deficit hyperactivity disorder (ADHD)) No (0) / Yes (1)

 NEPSY II: « Theory of mind »

Total Note Normal (0)

Lower normal (1)

Lower abnormal (2)

 NEPSY II : « Emotion recognition »

Total Note, Happy Error, Sad Error, Neutral Error, Fear Error, Anger Error, Disgust Error Normal (0)

Lower normal (1)

Lower abnormal (2)

I suspect a between-group analysis (AVH+ vs. AVH-) would be more appropriate. 

This is indeed an inter-group analysis between “AVH +” and “AVH -”. I modified the following sentence in the "Data analysis" paragraph: “For the main results, the correlation between the "AVH +" and "AVH -" groups for markers of social cognition (NEPSY II) and emotional markers (EED IV and BAVQ-R) in non-psychotic children and adolescents will be evaluated using correlation analyses”.

A correlational analysis also seems not best suited for the relationship between patients’ diagnosis (categorical I assume) and AVH (again, probably a categorical variable). 

Thank you for clarifying this point. I also edited the text to clarify what type of correlation will be achieved between two categorical variables: “For the correlation of categorical variables, the nullity test of a correlation coefficient will be used (or the Spearman method in the case where the variables do not follow a normal distribution).”. Apart from the test for the nullity of a correlation coefficient (which needed to be specified in the article), could you tell me which test is the most appropriate in your opinion?

Here as well, the authors state that correlational analysis is used because of the small sample size, but since this is a study protocol, they can just plan to include more participants if that’s what needed for an appropriate analysis. Overall, this section is most concerning and needs the most comprehensive rewrite.

As explained previously, the small population sample of the study is due to feasibility. The presence of auditory verbal hallucinations (AVH) in children and adolescents with post-traumatic stress disorder (PTSD) is a rarer symptomatology and more challenging to objectify in clinical interviews compared to the adult population. Despite our child and adolescent psychiatry department being a reference center for psychotrauma, the number of patients exhibiting this clinical profile remains limited. 

The study variables have been described earlier. This highlights at least 47 variables at T0 alone. Achieving a multiple regression analysis would require several hundred patients to be included in the study. Recruiting such a large number of patients is not feasible, both due to the rarity of the symptomatology and the chosen methodology, along with the available resources for conducting this study.

5. Hypothesis: In the discussion section the authors discuss a very interesting theory regarding the co-development of AVH and PTSD symptoms (“dys-sensory perception”), however as far as I can tell they plan no tests or analyses to investigate this model. I suggest the authors deduce specific predictions from this model that can be tested with their collected data.

Thank you for pointing out this element of the discussion which is also of particular interest to us. The hypothesis of our study in this article is that social cognition and emotional experiences are common markers to explain the presence and persistence of AVH in non-psychotic children and adolescents with PTSD. It is in this hypothesis that we propose the tests and analyzes to study this question. The “dys-sensory perception” model that we suggest is an opening to future transdisciplinary work with multisensory study perspectives and in association with other medical specialties (ENT, etc.)

Minor concerns:

1. Figure quality: The attached figures are not of sufficient quality for publication.

Thank you for this suggestion which allowed me to rework the quality of figure 1.

2. Hypotheses: The hypotheses in the introduction should be stated more clearly and in statistical terms.

Thank you for suggesting that we simply reformulate the hypothesis of the study which I add at the end of the introduction to the article: “The hypothesis of our study is the social cognitions and emotional markers are cognitive biases associated, or even correlated, with the presence of AVH in children and adolescents with PTSD. This study offers a prospective observation of these markers in the evolution of AVH and psychotic disorder in children and adolescents with PTSD”.

3. Data availability: The authors should add a data availability statement to the method section, detailing where their data will be made available.

The sentence was moved to the article and clarifi

---

## [Decision Letter · Decision Letter 1]

5 Feb 2024

PONE-D-23-25826R1Relationship between social cognition and emotional markers and acoustic-verbal hallucination in youth with post-traumatic stress disorder: protocol for a prospective, 2-year, longitudinal case-control studyPLOS ONE

Dear Dr. Dumas,

Thank you for submitting your manuscript to PLOS ONE. After careful consideration, we feel that it has merit but does not fully meet PLOS ONE’s publication criteria as it currently stands. Therefore, we invite you to submit a revised version of the manuscript that addresses the points raised during the review process.

We look forward to receiving your revised manuscript.

Kind regards,

Dirceu Henrique Paulo Mabunda, M.D.

Academic Editor

PLOS ONE

Journal Requirements:

**Additional Editor Comments:**

Please address the issues raised by reviewers.

Reviewers' comments:

Reviewer's Responses to Questions

**Comments to the Author**

1. Does the manuscript provide a valid rationale for the proposed study, with clearly identified and justified research questions?

Reviewer #1: Yes

Reviewer #2: Partly

Reviewer #3: Yes

2. Is the protocol technically sound and planned in a manner that will lead to a meaningful outcome and allow testing the stated hypotheses?

Reviewer #1: No

Reviewer #2: Partly

Reviewer #3: Yes

3. Is the methodology feasible and described in sufficient detail to allow the work to be replicable?

Reviewer #1: Yes

Reviewer #2: Yes

Reviewer #3: Yes

4. Have the authors described where all data underlying the findings will be made available when the study is complete?

Reviewer #1: Yes

Reviewer #2: Yes

Reviewer #3: Yes

5. Is the manuscript presented in an intelligible fashion and written in standard English?

Reviewer #1: Yes

Reviewer #2: Yes

Reviewer #3: Yes

6. Review Comments to the Author

You may also provide optional suggestions and comments to authors that they might find helpful in planning their study.

Reviewer #1: I thank the authors for providing a revision and responding politely and extensively to all my comments. The majority of concerns has been adressed. However, there still remain two central issues that I believe should be adressed:

1) Sample size calculation: The authors state that they were unable to calculate a required sample size. Nonetheless they need to include a sample size, so there is a clear criterion for when recruiting should be stopped. Indeed, the authors write that the "number of inclusions neccessary was reached on February 8, 2023". If no sample size is defined, this does not make sense. Apparently recruitment has stopped already, but as of now it is unclear why recruiting stopped if no sample size is determined. Since the authors specified their hypothesis I believe the appropriate way of calculcating a sample size would be to determine an effect size of interest (how large does the effect in question need to be to be considered relevant?) and then calculating the required number of participants to detect this effect. So, if the authors are able to implement this and adapt their recruiting strategy to the results I would think this is the best action to take. However, I suspect that circumstances might prevent further recruitung (e.g. lack of funding or recruitment possibilities). In that case, the authors should still calculate the required sample size and then state why they are unable to collect this sample transparently (which is something that happens all the time and thus should not be seen as a failure on the authors side but rather an accident of circumstances).

2) Data Analysis: The section on data analysis, while updated, is still not sufficiently detailled to allow for replication or meaningful outcomes. I think the issue starts with the hypothesis: "The hypothesis of our study is the social cognitions and emotional markers are cognitive biases associated, or even correlated, with the presence of AVH in children and adolescents with PTSD." First of, I don't understand the distinction between association and correlation drawn here. This should be clarified. Second, the "presence of AVH" is a categorical variable in this study, thus the correct analysis to test the hypothesis would a between-subject test (such as t-test) with "social cognitions" and "emotional markers" as outcomes. The hypothesis should state this clearly, e.g.: "The hypothesis of our study is that social cognitions and emotional markers are more/less strongly expressed in a AVH group than in a similar group without AVH". Ideally, this hypothesis already includeds the specific names of the variables from the specific measures applied. If the hypothesis is clearly formulated the section on data analysis can be written more clearly as well.

From my understanding of the collected data the second paragraph in the data analysis section ("A bivariate descriptive analysis will look for a significant difference between the AVH+ and AVH- groups. Continuous variables will be compared using the Student's test (or the non-parametric Wilcoxon test if the variables do not follow a normal distribution), and categorical variables with the chi-square test (or the non-parametric bilateral Fisher's exact test if the variables do not follow a normal distribution.)") seems a sufficient explanation of the testing procedure. The only neccessary addition would be a clear written out mention of all variables. For example: "Continuous variables tested with Student's test are: BAVQ-R, DES IV, etc.; Categorical variables assessed with the Wilcoxon test are: NEPSY II Theory of Mind, NEPSY II Emotion recognitoon, etc.". This should be a direct reflection of the hypothesis stated at the end of the introduction. As the authors state there are only two primary outcomes, I recommend only these two should be listed here. And if exploratory analyses are to be conducted they should be mentioned separately in a section on "exploratory analses".

This also applies to the section on secondary outcomes. To my understanding this would be a repetition of the first analysis, two years later? This should also be mentioned in the hypothesis so it is obvious what variables are tested specifically.

Minor points:

1) There is no need to repeated the procedure from the figures in detail in the "Procedure" section. Apparently that was a misunderstanding from my first review, for which I apologize.

2) The authors replied that they only assess AVH status via a question by the study evaluator. However, in the section "Assessment of group allocation" they still mention a "self-report questionnaire". Please explain what measure is used here.

3) Since the clinical trial registry belongs to the 14-7 programm this should be removed.

Reviewer #2: The authors have put in great effort to address the comments.

Spearman correlation is suitable for comparison of ordinal data or nonparametric (skewed) data. For comparison of two nominal data, Cramer’s V could be used.

Reviewer #3: I have had the opportunity to review the work "Relationship between social cognition and emotional markers and acoustic-verbal

hallucination in youth with post-traumatic stress disorder: protocol for a prospective, 2-

year, longitudinal case-control study", the modifications made by the authors have notably improved the previous version and have satisfactorily answered the questions and recommendations of the reviewers. I believe that the article could be ready for publication but, nevertheless, I I would like to make some recommendations in case you consider that it can be improved even further. Regarding the calculation of the effect size, in the event that there is no data from other research on the same characteristics, in addition to pointing this out as a limitation, other research could be used with similar populations (even though different variables have been measured). This is only a suggestion, due to the relevance that reporting the effect size may have. As for the statistical analyses, if the authors plan to obtain a small number of participants, the correlational analyzes they may not be the most appropriate and the authors should point out this as a limitation and suggest alternatives adjusted to the expected reality of the data. In this sense, I suggest that they add comparisons between groups of general linear models such as ANOVA, as an alternative.

7. PLOS authors have the option to publish the peer review history of their article (what does this mean?). If published, this will include your full peer review and any attached files.

Reviewer #1: **Yes: **Lukas Basedow

Reviewer #2: No

Reviewer #3: **Yes: **Juan A. Moriana

---

## [Author Response · Author response to Decision Letter 1]

20 Feb 2024

Reviewer #1: I thank the authors for providing a revision and responding politely and extensively to all my comments. The majority of concerns has been addressed. However, there still remain two central issues that I believe should be addressed:

1) Sample size calculation: The authors state that they were unable to calculate a required sample size. Nonetheless they need to include a sample size, so there is a clear criterion for when recruiting should be stopped. Indeed, the authors write that the "number of inclusions necessary was reached on February 8, 2023". If no sample size is defined, this does not make sense. Apparently recruitment has stopped already, but as of now it is unclear why recruiting stopped if no sample size is determined. Since the authors specified their hypothesis I believe the appropriate way of calculating a sample size would be to determine an effect size of interest (how large does the effect in question need to be to be considered relevant?) and then calculating the required number of participants to detect this effect. So, if the authors are able to implement this and adapt their recruiting strategy to the results I would think this is the best action to take. However, I suspect that circumstances might prevent further recruiting (e.g. lack of funding or recruitment possibilities). In that case, the authors should still calculate the required sample size and then state why they are unable to collect this sample transparently (which is something that happens all the time and thus should not be seen as a failure on the authors side but rather an accident of circumstances).

We thank you for your comment. To clarify, we propose to reinstate this information accompanied by an additional explanation which we hope is satisfactory.

We replace the "Simple size" paragraph with the following sentences: “The current literature data does not provide enough information for the calculation of the effect size and the required sample size. We were consequently unable to calculate a number of subjects necessary for the physalis study. We calculated a number of necessary subjects on the feasibility of recruiting and monitoring patients in the child psychiatry department and based on our previous work: a number of 15 patients per group, and a total number of 30 patients in total, were chosen. Considering a rate of loss of follow-up and missing data of around 30%, 20 patients are to be included per arm for a total of 40 patients”.

2) Data Analysis: The section on data analysis, while updated, is still not sufficiently detailed to allow for replication or meaningful outcomes. I think the issue starts with the hypothesis: "The hypothesis of our study is the social cognitions and emotional markers are cognitive biases associated, or even correlated, with the presence of AVH in children and adolescents with PTSD." First off, I don't understand the distinction between association and correlation drawn here. This should be clarified. 

Thank you for your vigilance. We agree with you and corrected accordingly. We removed the distinction between the terms "association" and "correlation".

Second, the "presence of AVH" is a categorical variable in this study, thus the correct analysis to test the hypothesis would a between-subject test (such as t-test) with "social cognitions" and "emotional markers" as outcomes. 

We agree with you. Concerning the comparison of the “AVH+” and “AVH-” groups, we made both Student’s test (continuous variables) and chi-square test (categorical variables).

As indicated in the appendix and to clarify this in the article, we changed the "data analysis" paragraph with the following sentences: “Continuous variables compared using the Student test (or the non-parametric Wilcoxon test if the variables do not follow a normal distribution) concern the emotional markers “DES IV” and “BAVQ-R”. Categorical variables compared using the chi-square test (or the non-parametric two-tailed Fisher exact test if the variables do not follow a normal distribution) concern the social cognition markers NEPSY II "Theory of Mind" and "Emotion Recognition "”.

The hypothesis should state this clearly, e.g.: "The hypothesis of our study is that social cognitions and emotional markers are more/less strongly expressed in a AVH group than in a similar group without AVH". Ideally, this hypothesis already included the specific names of the variables from the specific measures applied. If the hypothesis is clearly formulated the section on data analysis can be written more clearly as well.

We thank you for your suggestion to rewrite the study hypothesis. We made the following corrections: “The hypothesis of our study is that social cognitions and emotional markers are expressed differently in the presence or absence of AVH in children and adolescents with PTSD”.

From my understanding of the collected data the second paragraph in the data analysis section ("A bivariate descriptive analysis will look for a significant difference between the AVH+ and AVH- groups. Continuous variables will be compared using the Student's test (or the non-parametric Wilcoxon test if the variables do not follow a normal distribution), and categorical variables with the chi-square test (or the non-parametric bilateral Fisher's exact test if the variables do not follow a normal distribution.)") seems a sufficient explanation of the testing procedure. The only necessary addition would be a clear written out mention of all variables. For example: "Continuous variables tested with Student's test are: BAVQ-R, DES IV, etc.; Categorical variables assessed with the Wilcoxon test are: NEPSY II Theory of Mind, NEPSY II Emotion recognition, etc.". This should be a direct reflection of the hypothesis stated at the end of the introduction. 

Thank you for this suggestion to specify which variables are tested. As explained above, we modify the text with the following sentences: “Continuous variables compared using the Student test (or the non-parametric Wilcoxon test if the variables do not follow a normal distribution) concern the emotional markers “DES IV” and “BAVQ-R”. Categorical variables compared using the chi-square test (or the non-parametric two-tailed Fisher exact test if the variables do not follow a normal distribution) concern the social cognition markers NEPSY II "Theory of Mind" and "Emotion Recognition ".”

As the authors state there are only two primary outcomes, I recommend only these two should be listed here. And if exploratory analyses are to be conducted they should be mentioned separately in a section on "exploratory analyses".

This also applies to the section on secondary outcomes. To my understanding this would be a repetition of the first analysis, two years later? This should also be mentioned in the hypothesis so it is obvious what variables are tested specifically.

Thank you for advising us on the reorganization of the “Data analysis” paragraph which allows us to propose the following paragraph:

• “The main objective

In order to answer the study hypothesis, a bivariate descriptive analysis will look for a significant difference between the AVH+ and AVH- groups. Continuous variables compared using the Student test (or the non-parametric Wilcoxon test if the variables do not follow a normal distribution) concern the emotional markers “DES IV” and “BAVQ-R”. Categorical variables compared using the chi-square test (or the non-parametric two-tailed Fisher exact test if the variables do not follow a normal distribution) concern the social cognition markers NEPSY II "Theory of Mind" and "Emotion Recognition ". Since there are two primary outcomes, correcting the alpha risk by the Bonferroni method is necessary, with an adjusted alpha of 2.5% for each marker to control the overall familywise error rate.

• Exploratory analyses

1) A new statistical analysis will be carried out at two years to compare social cognition markers and emotional markers between “AVH+” and “AVH-” groups. 

2) The correlation between the "AVH +" and "AVH -" groups for social cognition (NEPSY II) and emotional markers (DES IV and BAVQ-R) in non-psychotic children and adolescents will be evaluated using correlation analyses. Pearson's correlation method was used to measure the association between two continuous variables in a small population sample. For the correlation of categorical variables, the nullity test of a correlation coefficient will be used (or the Spearman method in the case where the variables do not follow a normal distribution). Correlations between social cognition and emotional markers and the persistence of AVH in non-psychotic children and adolescents at two years will be reassessed.

3) The variables of the groups with or without persistence of AVH at two years will be compared and correlated to evaluate the evolution of the initial diagnosis of PTSD and the development of a psychotic disorder.”

This also allowed us to correct the writing of the study objectives: “The main objective of our study protocol is to compare social cognition markers and emotional markers according to the presence or absence of AVH in non-psychotic children and adolescents with PTSD. The secondary objectives of our study protocol are: 1) To compare social cognition markers and emotional markers according to the persistence at 2 years of AVH in non-psychotic children and adolescents with PTSD; 2) To evaluate the correlations between social cognition markers and emotional markers with AVH and with the persistence of AVH over two years; and 3) To evaluate social cognition and emotional markers associated with the evolution of PTSD and the development of psychotic disorder at two years.”

Minor points:

1) There is no need to repeated the procedure from the figures in detail in the "Procedure" section. Apparently that was a misunderstanding from my first review, for which I apologize.

Thank you for your response to my correction. So we delete the “Procedure” section.

2) The authors replied that they only assess AVH status via a question by the study evaluator. However, in the section "Assessment of group allocation" they still mention a "self-report questionnaire". Please explain what measure is used here.

Thank you for your vigilance. This is a question asked by the assessor and not a self-administered questionnaire. The term “self-report” has been removed from the text.

3) Since the clinical trial registry belongs to the 14-7 program this should be removed.

Thank you for this suggestion. The 14-7 study and the Physalis study are two different research protocols. But as explained in the paragraph " Ethics and dissemination", the Physalis study is an ancillary study to study 14-7 and thus benefits from the same ethical procedure and clinical trial registry.

Reviewer #2: The authors have put in great effort to address the comments.

Spearman correlation is suitable for comparison of ordinal data or nonparametric (skewed) data. For comparison of two nominal data, Cramer’s V could be used.

Thank you for your suggestion and we add this statistical analysis in the paragraph “Data analysis”: "A Cramer's V test in addition to the chi-square test will be carried out in order to measure the intensity of the relationship of nominal variables between two groups”.

Reviewer #3: I have had the opportunity to review the work "Relationship between social cognition and emotional markers and acoustic-verbal hallucination in youth with post-traumatic stress disorder: protocol for a prospective, 2-year, longitudinal case-control study", the modifications made by the authors have notably improved the previous version and have satisfactorily answered the questions and recommendations of the reviewers. I believe that the article could be ready for publication but, nevertheless, I would like to make some recommendations in case you consider that it can be improved even further. 

Regarding the calculation of the effect size, in the event that there is no data from other research on the same characteristics, in addition to pointing this out as a limitation, other research could be used with similar populations (even though different variables have been measured). This is only a suggestion, due to the relevance that reporting the effect size may have. 

Thank you for your comment, with which we fully agree. We based ourselves on our previous work studying a similar population to define this necessary number of subjects. We will also be rigorous in our search for studies of similar populations, in order to plan our future work.

As for the statistical analyses, if the authors plan to obtain a small number of participants, the correlational analyzes they may not be the most appropriate and the authors should point out this as a limitation and suggest alternatives adjusted to the expected reality of the data. 

Thank you for this suggestion which allows us to adjust the sentence which already mentioned this limit in the article "The choice of this method for a small sample size remains a limitation in the analysis of the results".

In this sense, I suggest that they add comparisons between groups of general linear models such as ANOVA, as an alternative.

Thanks for this suggestion. We would have very much liked to have performed an ANOVA analysis in this study if we had compared more than two groups. But in this study, comparisons are only made between two groups, "AVH+" and "AVH-", which excludes the use of ANOVA.

---

## [Decision Letter · Decision Letter 2]

16 Jun 2024

Relationship between social cognition and emotional markers and acoustic-verbal hallucination in youth with post-traumatic stress disorder: protocol for a prospective, 2-year, longitudinal case-control study

PONE-D-23-25826R2

Dear Dr Louis-Emilie Dumas

We’re pleased to inform you that your manuscript has been judged scientifically suitable for publication and will be formally accepted for publication once it meets all outstanding technical requirements.

Kind regards,

Dirceu Henrique Paulo Mabunda, M.D.

Academic Editor

PLOS ONE

Additional Editor Comments (optional):

Reviewers' comments:

Reviewer's Responses to Questions

**Comments to the Author**

1. Does the manuscript provide a valid rationale for the proposed study, with clearly identified and justified research questions?

Reviewer #3: Yes

Reviewer #4: Yes

2. Is the protocol technically sound and planned in a manner that will lead to a meaningful outcome and allow testing the stated hypotheses?

Reviewer #3: Yes

Reviewer #4: Yes

3. Is the methodology feasible and described in sufficient detail to allow the work to be replicable?

Reviewer #3: No

Reviewer #4: Yes

4. Have the authors described where all data underlying the findings will be made available when the study is complete?

Reviewer #3: Yes

Reviewer #4: Yes

5. Is the manuscript presented in an intelligible fashion and written in standard English?

Reviewer #3: Yes

Reviewer #4: Yes

6. Review Comments to the Author

You may also provide optional suggestions and comments to authors that they might find helpful in planning their study.

Reviewer #3: I believe that the authors have made the suggested changes and now it meets the requirements for publication.

Reviewer #4: I was able to read the study "Relationship between

social cognition and emotional markers and acoustic-verbal hallucination in youth with

post-traumatic stress disorder: protocol for a prospective, 2-year, longitudinal casecontrol

study"relatively late in the review process. The work by Dumas et al. provides interesting insights into the relationship of social cognition to AVH. Upon reading it became apparent that the authors have provided great care to work in the reviewer's comments. Thus we feel that this manuscript is ready for publication. However, I have two small recommendations:

1. I would advise to go over the manuscript one final time to improve wording and sentence structure - especially in the abstract.

2. I agree with the authors point of view that it is challenging to calculate needed sample sizes for power calculations if no precedents exist, however I would recommend providing a reference for the cited work that was used to do the actual sample size calculations in order to make the calculation replicable.

7. PLOS authors have the option to publish the peer review history of their article (what does this mean?). If published, this will include your full peer review and any attached files.

Reviewer #3: No

Reviewer #4: No

---

## [Editor Report · Acceptance letter]

23 Jun 2024

PONE-D-23-25826R2 

PLOS ONE

Dear Dr. Dumas, 

I'm pleased to inform you that your manuscript has been deemed suitable for publication in PLOS ONE. Congratulations! Your manuscript is now being handed over to our production team.

Kind regards, 

on behalf of

Dr. Dirceu Henrique Paulo Mabunda 

Academic Editor

PLOS ONE